# Cytology and High-Risk Human Papillomavirus Test for Cervical Cancer Screening Assessment

**DOI:** 10.3390/diagnostics12071748

**Published:** 2022-07-19

**Authors:** Frederik A. Stuebs, Martin C. Koch, Anna K. Dietl, Werner Adler, Carol Geppert, Arndt Hartmann, Antje Knöll, Matthias W. Beckmann, Grit Mehlhorn, Carla E. Schulmeyer, Paul Gass

**Affiliations:** 1Department of Gynecology and Obstetrics, Erlangen University Hospital, Comprehensive Cancer Center Erlangen–European Metropolitan Area of Nuremberg (CCC ER-EMN), Friedrich-Alexander-Universität Erlangen-Nürnberg, Universitaetsstrasse 21–23, 91054 Erlangen, Germany; anna.dietl@uk-erlangen.de (A.K.D.); matthias.beckmann@uk-erlangen.de (M.W.B.); carla.schulmeyer@uk-erlangen.de (C.E.S.); paul.gass@uk-erlangen.de (P.G.); 2Department of Gynecology and Obstetrics, Hospital ANregiomed Ansbach, Escherichstraße 1, 91522 Ansbach, Germany; martin.koch@anregiomed.de; 3Department of Medical Informatics, Biometry and Epidemiology, Friedrich-Alexander-Universität Erlangen-Nürnberg, Waldstrasse 6, 91054 Erlangen, Germany; werner.adler@fau.de; 4Institute of Pathology, Erlangen University Hospital, Comprehensive Cancer Center Erlangen–European Metropolitan Area of Nuremberg (CCC ER-EMN), Friedrich-Alexander-Universität Erlangen-Nürnberg, Krankenhausstrasse 8–10, 91054 Erlangen, Germany; carol.geppert@uk-erlangen.de (C.G.); arndt.hartmann@uk-erlangen.de (A.H.); grit.mehlhorn@uk-erlangen.de (G.M.); 5Institute of Clinical and Molecular Virology, Erlangen University Hospital, Friedrich-Alexander-Universität Erlangen-Nürnberg, Schlossgarten 4, 91054 Erlangen, Germany; antje.knoell@uk-erlangen.de; 6Gynecology Consultancy Practice, German Cancer Society [DKG] and Committee on Cervical Pathology and Colposcopy [AG-CPC] Certified Gynaecological Dysplasia Consultancy Practice, Frauenarztpraxis Erlangen, Neustädter Kirchenplatz 1a, 91054 Erlangen, Germany

**Keywords:** high-grade squamous lesion (HSIL), cervical dyplasia, cervical cancer, Pap smear, human papillomavirus, screening for cervical cancer, colposcopy

## Abstract

Background: A new nationwide screening strategy was implemented in Germany in January 2020. No data are available for women referred to certified dysplasia units for secondary clarification after primary diagnosis by a local physician. We therefore investigated combined testing with Papanicolaou smears and high-risk human papillomavirus (hrHPV) and compared the data with the final histological findings. Methods: Between January 2015 and October 2020, all referred women who underwent colposcopy of the uterine cervix in our certified dysplasia unit were included. Cytology findings were classified using the Munich III nomenclature. Results: A total of 3588 colposcopies were performed in 3118 women, along with Pap smear and hrHPV co-testing, followed by histology. Women with Pap II-p (ASC-US) and a positive hrHPV co-test had a 22.4% risk for cervical intraepithelial neoplasia (CIN) 3/high-grade squamous intraepithelial lesion (HSIL). The risk of CIN 3/HSIL was 83.8% in women with Pap IVa-p (HSIL) and a positive hrHPV co-test. A positive hrHPV co-test increased the risk for HSIL+ (OR 5.942; 95% CI, 4.617 to 7.649; *p* < 0.001) as compared to a negative hrHPV co-test. Conclusions: The accuracy of Pap smears is comparable with the screening results. A positive hrHPV test increases the risk for HSIL+ fivefold. Colposcopy is necessary to diagnose HSIL+ correctly.

## 1. Introduction

Cervical cancer is the fourth most common cause of gynecological cancer–related death among women in Germany. Since the introduction of a nationwide screening program in 1971, the incidence and mortality have been declining over three decades. During the past 15 years, however, the incidence of cervical cancer has been stagnating at a low level, despite advances in diagnosis and therapy [1,2,3]. Invasive cervical cancer is caused by a persistent infection with human papillomavirus (HPV) [4,5,6]. HPV is a biological carcinogen, and high-risk HPV (hrHPV) types are detected in up to 99.7% of cervical cancers [4,7]. In large epidemiological studies, hrHPV has been detected in 85–93% of women diagnosed with cervical cancer [5,8,9]. The most common genotypes are hrHPV types 16 and 18 (HPV-16, HPV-18), which are found in almost 70% of cervical cancers, in 50–90% of cervical intraepithelial neoplasia 2–3 (CIN 2–3)/high-grade squamous lesions (HSILs), and in 25% of CIN 1/low-grade squamous intraepithelial lesions (LSILs) [5,9,10].

A new organized screening program was implemented in Germany in January 2020. Women between 20 and 34 years of age are continuing to have annual Pap smears, while women over the age of 34 receive a co-test comprising a Pap smear in combination with a hrHPV test every 3 years. All women aged 20–65 will be invited for testing by their health-insurance providers every 5 years [11,12,13,14,15,16,17].

Worldwide, the nomenclature most commonly used for Pap smears is the Bethesda classification [18], but the Munich III classification is routinely used in Germany [19,20]. HPV testing is more sensitive but less specific than cytology for detecting CIN, as hrHPV is present in about 10% of cervical samples without morphologic changes. Women who receive a single hrHPV test have a significantly reduced relative risk of mortality from cervical cancer in comparison with those who have cytology alone (RR 0.59; 95% CI, 0.39 to 0.91) [21]. hrHPV-based screening also shows a higher detection rate for CIN 3+ than cytology-based screening, even if women only participate once (RR 1.23; 95% CI, 0.91 to 1.67) [22,23,24,25,26]. In the second round of screening with cytology, only the rate of detected CIN 3+ is greater (RR 0.52; 95% CI, 0.35 to 0.76). The risk of cervical cancer is also lower with hrHPV-based screening in comparison with cytology-based screening (RR 0.29; 95% CI, 0.11 to 0.73). This implies that hrHPV-based screening provides protection [22,23,24,25,26,27]. In women aged 25–34, however, hrHPV screening leads to overdiagnosis of regressive CIN 2 [22]. 

There are several reports in the literature about the positive predictive value (PPV) of cytology and hrHPV-based screening tests. Luyten et al. found that Pap I (negative for intraepithelial lesion or malignancy, NILM) with a negative hrHPV test is associated with a 0.005% likelihood (95% CI, 0.0001% to 0.03%) for CIN 3+ and Pap I (NILM), while a positive hrHPV test had a likelihood of 9.2% (95% CI, 7.4% to 10.9%) [28]. Similar results have been described for Pap II-p (atypical squamous cells of undetermined significance, ASC-US) with a negative versus a positive hrHPV result (0.43% vs. 6.8%, 95% CI, 5% to 33%) [29,30,31]. For Pap IIID1 (LSIL), the likelihood for CIN 3+ with a negative hrHPV result is 2.0%, in comparison with up to 46% with a positive hrHPV result [30,31,32].

For practical reasons, the positive and negative predictive values (PPV and NPV) both need to be high enough in screening tests for detection of cervical cancer [33,34]. The German guidelines on the prevention of cervical cancer recommends a colposcopy if the post-test likelihood for CIN 3+ is greater than 10% [21].

To the best of our knowledge, no published recommendations are available regarding the correct procedure in the case of a normal colposcopy with suspicious co-test results of the Pap and hrHPV tests in Germany. We therefore compared the results of the hrHPV and Pap co-test with the histological findings in each case seen in our certified dysplasia unit.

## 2. Materials and Methods

Dysplasia units have been established nationwide in Germany in accordance with the certification system of the German Cancer Society (*Deutsche Krebsgesellschaft e.V.*, DKG), the German Society for Gynecology and Obstetrics (*Deutsche Gesellschaft für Gynäkologie und Geburtshilfe e.V.*, DGGG), the Working Group on Gynecological Oncology (*Arbeitsgemeinschaft Gynäkologische Onkologie*, AGO), and the Working Group on Cervical Pathology and Colposcopy (*Arbeitsgemeinschaft für Zervixpathologie und Kolposkopie*, AGCPC). Dysplasia units cooperate with gynecological cancer centers in order to integrate in-patient health-care facilities [35]. Between January 2015 and October 2020, Pap smears and HPV samples from the cervix of the uterus were taken during 3588 colposcopies in 3118 women in the certified Dysplasia Unit at Erlangen University hospital. Abnormal cervical cytology findings were the most common reason for women being referred to the Dysplasia Unit, but in this study we also included women who were referred for other reasons, such as Lichen sclerosus or dysplasia of the vagina und vulva to generate a large set of women to achieve significant results. Before the first of January 2020 an opportunistic screening was taking place in Germany. Women were not invited for secondary prevention, but had the opportunity to see their gynecologist for an annual free-of-charge Pap smear from the age of 20 years [36].

A conventional Pap smear of the cervix (using the Munich III nomenclature), a test for hrHPV, and the application of 5% acetic acid to the cervix represent the standard of care in the unit, and are performed on all women referred to us in the described order. Only PAP smears, hrHPV-tests and histology obtained at the same visit as colposcopy was included in this study. Between 2015 and October 2018, the Hybrid Capture^®^ 2 test (HC2) was used to detect hrHPV (n = 2459). The HC2 test has a sensitivity of 94.1% (95% CI, 84.9 to 98.1) and a specificity of 87.9% (95% CI, 86.9 to 88.9) for detecting CIN 2+ [37]. The HC2 test is a molecular hybridization and sandwich capture assay that detects the 12 hrHPV types and type 68, classified as probably carcinogenic to humans (Group 2A) by IARC. The length of the full length of the HPV genome is the target of this test [38]^.^ Since November 2018 hrHPV is performed with the Abbott RealTime high-risk HPV assay on an Abbott *m*2000sp instrument (n = 1129). The Abbott RealTime high-risk HPV assay is a qualitive in vitro test for the detection of DNA [39]. This assay separately detects HPV-16, HPV-18, and a pool of 12 additional hrHPV types (HPV-31, -33, -35, -39, -45, -51, -52, -56, -58, -59, -66, and -68). The test has a sensitivity of 98.5% (95% CI, 91.0 to 99.9) and a specificity of 89.7% (95% CI, 88.8 to 90.6) [37].

The colposcopies were performed in standardized conditions using a Zeiss KSK 150 FC colposcope. General assessment was in accordance with the 2011 International Federation for Cervical Pathology and Colposcopy (IFCPC) colposcopic terminology of the cervix [36]. The results of the cervical Pap smears were classified in accordance with the Munich III nomenclature [19,20]. For clinicians not familiar with Munich III nomenclature the corresponding Bethesda classification is shown in brackets [18].

If there is a major finding or a lesion that is suspicious for invasion, a colposcopy-directed biopsy has to be taken from the most suspicious part of the lesion, using biopsy forceps (Seidl Biopsy Forceps ER076R; Aesculap AG, Tuttlingen, Germany).

During the period of this retrospective analysis, the team in the Dysplasia Unit consisted of five colposcopists with various degrees of clinical experience and training. All data—such as colposcopic findings, Pap smear and hrHPV test results, histological outcomes, number of biopsies, type of transformation zone, and epidemiological outcomes—were recorded in a database for further research.

### Statistics

For statistical analysis, the different Pap grades were subdivided into four different groups (testing for HPV was carried out for all Pap smears) (See Table 1).

The histological findings were subdivided into two groups: LSIL− (benign and CIN 1/LSIL) versus HSIL+ (CIN 2/HSIL, CIN 3/HSIL; AIS/HSIL; carcinoma).

To evaluate the diagnostic benefit of hrHPV testing, two logistic regression models with dichotomous histology (LSIL− vs. HSIL+) were calculated as dependent variables, and Pap as an independent variable. As there were 3588 observations from 3118 women in the dataset, so that not all observations are truly independent, generalized estimating equations (GEEs), which take the dependency structure in the data into account, were used to estimate parameters and standard errors in the logistic regression models. In one of the models, HPV was included as an additional independent variable. Using these models, the probabilities for HSIL+ were calculated and receiver operating characteristic (ROC) analysis was completed. The areas under the ROC curve (AUC) were calculated and compared using the DeLong test. To verify that the results produced by this test were not biased due to the existing dependency between some of the observations, a somewhat more complex bootstrap analysis was performed, which led to the same results and is therefore not presented in the text here (see the Appendix A).

The significance level was set to 0.05. All statistical calculations were completed using the statistical programming language R, version 4.0.3 [40].

## 3. Results

Between January 2015 and October 2020, a total of 17.686 colposcopies were performed in the Dysplasia Unit at Erlangen University Hospital. 3118 women underwent colposcopy followed by histology or surgery (see Figure 1). The women’s mean age was 35.9 years (standard deviation 10.3 years). A total of 6948 colposcopies with Pap smears and hrHPV co-testing were performed during that period, and 3588 of the colposcopies were followed by histology (51.6%). In 297 colposcopies, the women were diagnosed with Pap I (NILM); 257 of these patients (86.5%) had negative hrHPV co-tests (Table 2).

In 246 colposcopies, the women’s smear results showed Pap II-a (NILM); 192 of these women (78%) were negative for hrHPV. Eighty-eight of the colposcopies led to benign histology findings regardless of the hrHPV co-test; 57% had CIN 1/LSIL in the histological findings. In 17 colposcopies, women with Pap II-a (NILM) had histology results showing HSIL (CIN 2 or CIN 3). One woman with Pap II-a (NILM) was diagnosed with cervical cancer (Table 2).

In 144 cases, the women were diagnosed with Pap II-p (ASC-US). Women with Pap II-p (ASC-US) and a positive hrHPV co-test had a 22.4% likelihood of having CIN 3/HSIL, in comparison with a 1.1% likelihood with Pap II-p and a negative hrHPV co-test (Table 2).

In 663 colposcopies, the women were diagnosed with Pap IIID1 (LSIL), and the results of hrHPV testing were equally balanced between negative and positive findings. Women with Pap IIID1 (LSIL) and a positive hrHPV co-test have a 14.1% risk of having CIN 3/HSIL, but only a 3.6% risk when the hrHPV co-test is negative (Table 2).

In 508 cases, the women were diagnosed with Pap IIID2 (HSIL). 78.9% had a positive hrHPV testing. 39.2% of cases of Pap IIID2 and a positive hrHPV testing had CIN 3/HSIL results, followed by 136 cases of CIN 2/HSIL (33.9%) (Table 2).

Among 258 women with Pap III-p (ASC-H), 168 were positive for hrHPV (65.1%). Among women with Pap III-p (ASC-H) and a positive hrHPV co-test, 60.7% were diagnosed with CIN 3/HSIL, but only 13.3% had negative hrHPV co-tests. Women with negative hrHPV co-tests were more likely to be diagnosed with benign histology or CIN 1/LSIL in comparison with women who had a positive test (Table 2).

Among women with Pap IVa-p (HSIL), 92.1% (n = 1032) also tested positive for hrHPV; 83.8% of these women were diagnosed with CIN 3/HSIL (n = 865), but only 60.2% of those with negative hrHPV co-tests were diagnosed with CIN 3/HSIL (Table 2).

Among women with Pap V-p (squamous cell carcinoma), 65.1% were diagnosed with cervical cancer; 94.2% of those with Pap V-p (squamous cell carcinoma) were positive for hrHPV (Table 2). Table 3 and Table 4 present the data relative to age (<35 vs. >35 years). This shows that the results for PAP IIID1 (LSIL), IIID2 (HSIL), III-p (ASC-H) and IVa-p (HSIL) are fairly balanced between the two age groups.

In the GEE model with dichotomous histology (LSIL− vs. HSIL+), an odds ratio (OR) of 34.27 (95% CI, 24.629 to 47.671; *p* < 0.001) was observed for Pap HSIL+. The risk for histology with HSIL+ was nine times increased for PAP category III (Pap-unspecific findings; OR 9.24; 95% CI, 6.346 to 13.46; *p* < 0.001) (Table 5). A positive hrHPV co-test increases the risk for HSIL+ histology by a factor of five (OR 5.07; 95% CI, 4.068 to 6.329; *p* < 0.001). The ROC analysis for the two different GEE models showed an increased AUC for the model with a positive HPV in comparison with the model with a negative HPV (0.887 vs. 0.858) (Figure 2). The increased AUC for the model with a positive hrHPV co-test shows the increased likelihood of a positive hrHPV to diagnose HSIL+.

## 4. Discussion

The majority of the patients (n = 1120) were diagnosed with Pap IVa-p (HSIL); 92.1% of these patients also had positive tests for hrHPV, and 83.8% were diagnosed with CIN 3/HSIL. In 338 of 663 patients (58.5%) with Pap IIID1 (LSIL), the histology was CIN 1/LSIL. Among women diagnosed with Pap II-p (ASC-US) in combination with a positive hrHPV test, the likelihood for CIN 3/HSIL was 22.4%, in comparison with 1.1% with a negative hrHPV test.

A new screening program was implemented in Germany in January 2020. Women between 20 and 34 still receive an annual Pap smear, but women over the age of 35 have a co-test every 3 years [12]. There are clear recommendations on how to handle the different results of Pap and hrHPV co-testing in the different age groups. Women with Pap IV (HSIL) or Pap V (carcinoma) must be referred to a dysplasia unit immediately, regardless of age and hrHPV status. Women with Pap III-p (ASC-H), III-g (AGC endocervical favoring neoplasia) or IIID2 (HSIL) are to be referred to a dysplasia unit within 3 months, regardless of age and hrHPV status. Depending on age, hrHPV status, and cytology findings, women with other Pap results are to be referred to a dysplasia unit within 3 months or receive another Pap test alone or with a hrHPV co-test within 6–12 months [12]. Despite clear instructions how to manage women with suspicious screening results, there are no instructions about the procedure after women have been referred to a dysplasia unit in Germany. When a histological sample has been obtained during the colposcopy examination, the subsequent procedure depends on the histology results. Women with HSIL+ need a surgical intervention, whereas women with LSIL can undergo observation [2]. The situation is more difficult if the Pap smear and hrHPV co-test are suspicious for a high-grade lesion but the colposcopy and/or biopsy results are unsuspicious.

In the annual statistics on cervical screening in Germany, data for 15,124,043 women are available for 2015 [41]. Among all the women screened, 95.9% had unsuspicious test results and were referred back to screening; 84.8% of women with Pap I (NILM) had benign histological results. The rates for CIN 3/HSIL, squamous cancer of the cervix, adenocarcinoma of the cervix, and endometrial cancer were 4.5%, 0.5%, 0.7%, and 2.4%, respectively. In the present study, only one woman with Pap I (NILM) was diagnosed with CIN 3/HSIL (0.3%); she tested positive for hrHPV. Taking only women with positive hrHPV co-tests into account, the rate increases to 2.5%. This is similar to other reports in the literature [28,29].

For Pap II-a (NILM) in the annual statistics, benign results were found in 68.8%. The rate of CIN 3/HSIL increased to 8.18% in comparison with women with Pap I. In the group of women included in the present study, the rate of CIN 3/HSIL more than doubled when the hrHPV test was positive in comparison with women with Pap I: 5.6% versus 2.5%.

Women with Pap II-p (ASC-US) in the annual statistics had a 12.5% risk of being diagnosed with CIN 3/HSIL, and only 54.8% had benign histology findings. Among women with Pap II-p (ASC-US) and a positive hrHPV co-test, 26.5% had benign histological findings in the group of women included in the present study. The rate increased to 55.8% when the hrHPV co-test was negative. The difference is even larger with the histologic results of CIN 3/HSIL: 22.4% versus 1.1%, respectively. In another study, the risk for CIN 3/HSIL+ was reported as 0.43 in women with Pap II-p (ASC-US) and a negative hrHPV co-test, but there was a risk of 6.8% in case of a positive hrHPV co-test [29]. The increased rate of CIN 3/HSIL in the group of women included in the present study is due to the colposcopic results. The CIN 3/HSIL lesions were not detected with cytology, but were seen at colposcopy.

In another study from the U.S., the risk for women with Pap IIID1 (LSIL) and a negative hrHPV co-test was 2.0% for a diagnosis of CIN 3/HSIL+, but the risk increased to 6.1% for women with a positive hrHPV co-test [32]. The risk of CIN 3/HSIL+ is further increased to 31.7% in women over the age of 30 in a German Study [31]. Among the women in the present study who were diagnosed with Pap IIID1 (LSIL), the risk for CIN 3/HSIL was 14.1% with a positive hrHPV co-test in comparison with 3.6% with a negative hrHPV co-test. However, the majority of women (58.5%) with Pap IIID1 (LSIL) were diagnosed with CIN 1/LSIL. Only 37.9% of women diagnosed with Pap IIID1 during screening had a diagnosis of CIN 1/LSIL. The rate of CIN 3/HSIL was 16.96% [42].

In the annual statistics on cervical screening in Germany, 25,161 women were diagnosed with Pap IVa-p (HSIL), representing 0.166% of all women. Among these women, 82.2% had CIN 3/HSIL, 8.6% had CIN 2/HSIL, and 2.5% had squamous cervical cancer [41]. These data are similar to our own. Women diagnosed with Pap IVa-p (HSIL) and with a positive hrHPV co-test had CIN 3/HSIL in 83.8% and cervical cancer in 2.5% of cases. In the literature, women with Pap V-p (squamous cell carcinoma) have an over 60% risk for squamous cervical cancer and an over 20% risk for CIN 3/HSIL [42]. These data are in line with our own. The present results show that even in a highly specialized unit, there is still some interobserver variability in the diagnosis of Pap smears. Although highly trained personnel evaluate the Pap smears and the colposcopic findings of the examination are documented, the results of the Pap smears are comparable to the results in the screening program. In some cases, the Pap smears and histologic samples were evaluated by the same pathologist, representing another potential bias in the results of the present study. A high percentage of CIN 3/HSIL was observed along with some Pap smears (e.g., Pap II-p, Pap IIID1, and Pap IIID2), even though the Pap smears themselves suggested a much less severe lesion. Biopsies were nevertheless taken because the examiner saw the need to rule out HSIL or invasion. This aspect further underlines the importance of adequate colposcopy.

The sensitivity of HPV testing for detecting HSIL+ is greater than with cytology-based tests, but the specificity is lower [43]. This effect was also seen in the present group of women. The sensitivity increased from 0.807 without HPV to 0.880 with HPV as an independent predictor. This shows that hrHPV co-testing is also valuable in this specialized setting for detecting HSIL lesions, and it should therefore be performed routinely in dysplasia units. A positive HPV test is also more reproducible and has a better predictive value than cytology alone [21,44]. Women with a positive hrHPV test result should be monitored closely. This can be conducted in a specialized dysplasia unit or by a general gynecologist, depending on the Pap grading of the examination. Women with Pap II-p or greater in combination with a positive hrHPV test should be seen in a certified dysplasia unit.

No recommendations are available on the further procedure for women who have been seen in a dysplasia unit with a suspicious Pap result and/or a positive hrHPV test without a histology obtained. More research and data is needed to come up with common nationwide recommendations that can be provided for women affected with suspicious Pap-smear and hrHPV.

### Strengths and Limitations

This study includes a large set of women who were seen in a certified dysplasia unit. It is a highly selected group of patients, the majority of whom were referred to the dysplasia unit due to suspicious cytology findings. However, the group also included women who were referred to the dysplasia unit for other reasons, such as dysplasia of the vulva or lichen sclerosus. The cytological and histological findings were analyzed in the same department, in some cases by the same examiner. The cytologists were aware of the colposcopic appearance and therefore knew whether there was a suspicious lesion. This may have influenced the results. No information was available regarding the HPV vaccination status of the women referred. The team treating the women is comparatively small, and the physicians are highly specialized in treating HSIL and cervical cancer.

## 5. Conclusions

A new screening program was introduced in Germany in January 2020. Women over the age of 35 now receive a cytological examination and hrHPV co-testing every 3 years. An increased rate of accuracy was observed for Pap IIID1 (LSIL) and Pap V-p (carcinoma) in our certified Dysplasia unit in comparison with the results reported from screening for general public. An increase in accuracy was not seen in women diagnosed with Pap IIID2 (HSIL) or Pap IVa-p, for example, despite the fact that colposcopy was performed and the cytologist was aware of the colposcopic appearance. As expected, a positive hrHPV co-test increases the risk for histologic HSIL+ in women referred to a dysplasia unit. Carrying out this co-test is an essential part of the diagnostic routine in a dysplasia unit. The procedure is straightforward for women with histology findings, but no guidelines are available for the further procedure in women with suspicious cytology results and a hrHPV co-test. More data and research are needed on women who have been referred to a dysplasia unit with unclear results, with an awareness that each woman’s individual risk profile will influence the subsequent procedure.

## Figures and Tables

**Figure 1 diagnostics-12-01748-f001:**
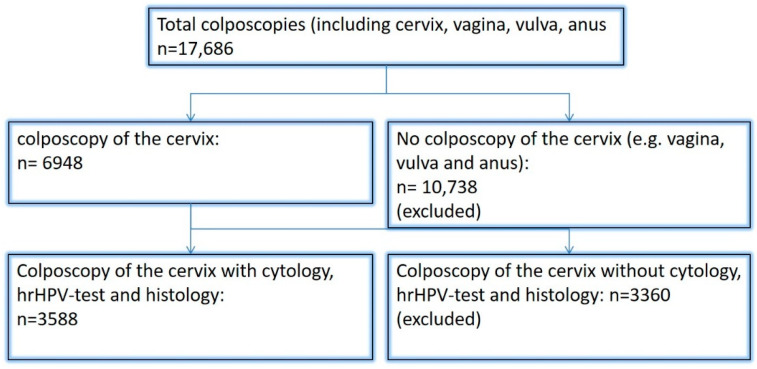
Flowchart.

**Figure 2 diagnostics-12-01748-f002:**
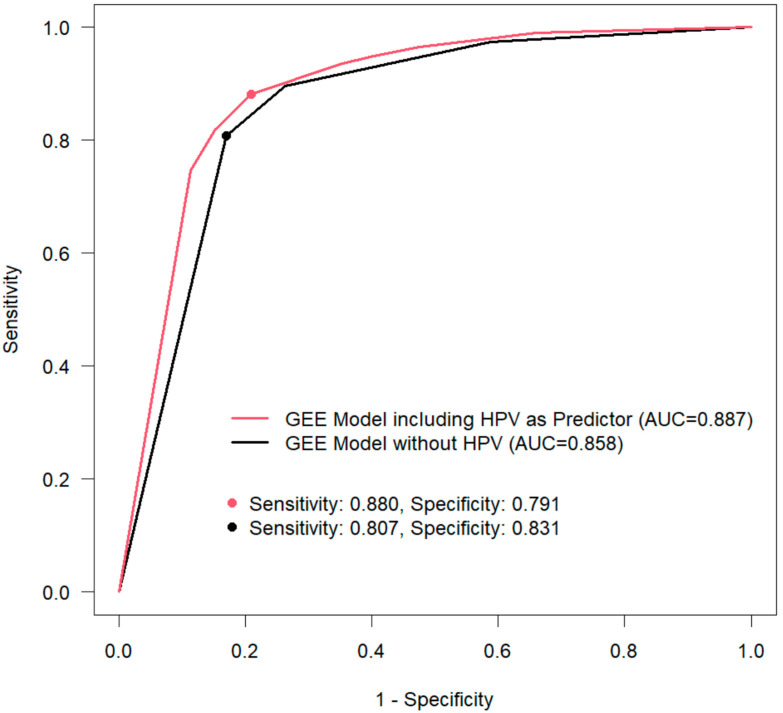
The generalized estimating equation (GEE) model, with and without HPV as a predictor. AUC, area under the curve.

**Table 1 diagnostics-12-01748-t001:** Subdivisions of PAP-Groups for statistical analysis (NILM:Negative for intraepithelial lesions or malignancy; AGC atypical glandular cells; NOS: not otherwise specified; ASC-US: Atypical squamous cells of undetermined significance; HSIL: High-grade squamous intraepithelial lesion; AIS: Adenocarcinoma in situ; ASC-H: atypical glandular cells of undetermined significance cannot exclude HSIL).

Subdivision	Munich III	Bethesda
Benign	I;	NILM
	II-a;	NILM
	II-g	AGC endocervical NOS
	II-p	ASC-US
LSIL	IIID1	LSIL
HSIL+	IIID2	HSIL
	IVa-p	HSIL
	IVa-g	AIS
	IVb-p	HSIL with features suspicious for invasion
	IVb-g	AIS with features suspicious for invasion
	V-e	endometrial adenocarcinoma
	V-g	endocervical adenocarcinoma
	V-p	squamous cell carcinoma
	V-x	other malignant neoplasms
Unspecific	III-e	AGC endometrial
	III-g	AGC endocervical favoring neoplasia
	III-p	ASC-H
	III-x	AGC favoring neoplasia

**Table 2 diagnostics-12-01748-t002:** Pap and HPV in Combination with Histology for the whole set of women.

Pap Smears with Histology (n = 3588) (3118 Women)	Bethesda System	hrHPV-Positive (n = 2369)	hrHPV-Negative (n = 1219)	Benign (n = 597)	CIN 1/LSIL (n = 964)	CIN 2/HSIL (n = 442)	CIN 3/AIS/HSIL (n = 1452)	Carcinoma (n = 133)
0 (n = 10)	Unsatisfactory for evaluation	5 (50%)		1 (20%)	3 (60%)	0	1 (20%)	0
0	Unsatisfactory for evaluation		5 (50%)	3 (60%)	2 (40%)	0	0	0
I (n = 297)	NILM	40 (13.5%)		25 (62.5%)	12 (30%)	2 (5%)	1 (2.5%)	0
I	NILM		257 (86.5%)	165 (64.2%)	89 (34.6%)	3 (1.2%)	0	0
II-a (n = 246)	NILM	54 (21.9%)		16 (29.6%)	29 (53.7%)	6 (11.1%)	3 (5.6%)	0
II-a	NILM		192 (78.0%)	72 (37.5%)	111 (57.8%)	4 (2.1%)	4 (2.1%)	1 (0.5%)
II-g (n = 6)	AGC endocervical NOS	2 (33.3%)		0	0	0	1 (50%)	1 (50%)
II-g	AGC endocervical NOS		4 (66.6%)	2 (50%)	2 (50%)	0	0	0
II-p (n = 144)	ASC-US	49 (34.1%)		13 (26.5%)	18 (36.7%)	7 (14.3%)	11 (22.4%)	0
II-p	ASC-US		95 (65.9%)	53 (55.8%)	33 (34.7%)	8 (8.4%)	1 (1.1%)	0
IIID1 (n = 663)	LSIL	325 (49.1%)		42 (12.9%)	175 (53.8%)	62 (19.1%)	46 (14.1%)	0
IIID1	LSIL		338 (50.9%)	74 (21.9%)	213 (63.0%)	39 (11.5%)	12 (3.6%)	0
IIID2 (n = 508)	HSIL	401 (78.9%)		27 (6.7%)	79 (19.7%)	136 (33.9%)	157 (39.2%)	2 (0.5%)
IIID2	HSIL		107 (21.1%)	17 (15.9%)	46 (43%)	34 (31.8%)	10 (9.3%)	0
III-e (n = 1)	AGC endometrial	0		0	0	0	0	0
III-e	AGC endometrial		1 (100%)	1 (100%)	0	0	0	0
III-g (n = 63)	AGC endocervical favoring neoplasia	42 (66.7%)		7 (16.6%)	9 (21.4%)	7 (16.6%)	14 (33.3%)	5 (11.9%)
III-g	AGC endocervical favoring neoplasia		21 (33.3%)	11 (52.4%)	7 (33.3%)	3 (14.3%)	0	0
III-p (n = 258)	ASC-H	168 (65.1%)		20 (11.9%)	25 (14.9%)	18 (10.7%)	102 (60.7%)	3 (1.8%)
III-p	ASC-H		90 (34.9%)	13 (14.4%)	49 (54.4%)	16 (17.8%)	12 (13.3%)	0
III-x (n = 3)	AGC favoring neoplasia	0		0	0	0	0	0
III-x	AGC favoring neoplasia		3 (100%)	2 (66.6%)	1 (33.3%)	0	0	0
IVa-p (n = 1120)	HSIL	1032 (92.1%)		20 (1.9%)	40 (3.9%)	81 (7.8%)	865 (83.8%)	26 (2.5%)
IVa-p	HSIL		88 (7.9%)	9 (10.2%)	13 (14.8%)	10 (11.4%)	53 (60.2%)	3 (3.4%)
IVa-g (n = 17)	AIS	15 (88.2%)		2 (11.8%)	2 (11.8%)	0	9 (52.9%)	2 (11.8%)
IVa-g	AIS		2 (11.8%)	1 (50%)	1 (50%)			
IVb-p (n = 142)	HSIL with features suspicious for invasion	132 (92.9%)		1 (0.8%)	4 (3.0%)	5 (3.8%)	111 (84.1%)	11 (8.3%)
IVb-p	HSIL with features suspicious for invasion		10 (7.0%)	0	1 (10%)	0	7 (70%)	2 (20%)
IVb-g (n = 9)	AIS with features suspicious for invasion	8 (88.9%)		0	0	0	5 (62.5%)	3 (37.5%)
IVb-g	AIS with features suspicious for invasion		1 (11.1%)	0	0	0	1 (100%)	0
V-p (n = 86)	Squamous cell carcinoma	81 (94.2%)		0	0	1 (1.2%)	24 (27.9%)	56 (65.1%)
V-p	Squamous cell carcinoma		5 (5.8%)	0	0	0	1 (20%)	4 (80%)
V-g (n = 12)	Endocervical adenocarcinoma	12 (100%)		0	0	0	1 (8.3%)	11 (91.7)
V-g	Endocervical adenocarcinoma		0	0	0	0	0	0
V-e (n = 1)	Endometrial adenocarcinoma	1 (100%)		0	0	0	0	1 (100%)
V-e	Endometrial adenocarcinoma		0	0	0	0	0	0
V-x (n = 2)	Other malignant neoplasms	2 (100%)		0	0	0	0	2 (100%)
V-x	Other malignant neoplasms		0	0	0	0	0	0

Abbreviations: AGC, atypical glandular cells; AIS, adenocarcinoma in situ; ASC-H, atypical squamous cells, HSIL not excluded; ASC-US, atypical squamous cells of undetermined significance; CIN, cervical intraepithelial neoplasia; hrHPV, high-risk human papillomavirus; HSIL, high-grade squamous intraepithelial lesion; LSIL, low-grade squamous intraepithelial lesion; NILM, negative for intraepithelial lesion or malignancy; NOS, not otherwise specified.

**Table 3 diagnostics-12-01748-t003:** Results for Pap IIID1, IIID2, III-p, and IVa-p in comparison the histologic results in women aged < 35.

Pap Smear (n = 1409)	hrHPV-Positive (n = 1137)	hrHPV-Negative (n = 272)	Benign (n = 108)	CIN 1 (n = 311)	CIN 2 (n = 222)	CIN 3/AIS (n = 753)	Carcinoma (n = 15)
Pap IIID1 (n = 347)	195 (56.2%)		28 (14.6%)	103 (52.8%)	37 (19.0%)	27 (13.8%)	0
Pap IIID1		152 (43.8%)	25 (16.4%)	93 (61.2%)	25 (16.4%)	9 (5.9%)	0
Pap IIID2 (n = 286)	239 (83.6%)		14 (5.6%)	58 (24.3%)	80 (33.5%)	85 (35.6%)	2 (0.8%)
Pap IIID2		47 (16.4%)	12 (25.5%)	11 (23.4%)	18 (38.3%)	6 (12.8%)	0
Pap III-p (n = 100)	77 (77%)		5 (6.5%)	9 (11.7%)	7 (9.1%)	55 (71.4%)	1 (1.3%)
Pap III-p		23 (23%)	3 (13.0%)	9 (39.1%)	6 (26.1%)	5 (21.7%)	0
Pap IVa-p (n = 676)	626 (92.6%)		14 (2.2%)	19 (3.0%)	45 (7.2%)	538 (85.9%)	10 (1.6%)
Pap IVa-p		50 (7.4%)	7 (14.0%)	9 (18.0%)	4 (8.0%)	28 (58%)	2 (4.0%)

Abbreviations: AIS, adenocarcinoma in situ; CIN, cervical intraepithelial neoplasia; hrHPV, high-risk human papillomavirus. Comparison of Munich III and Bethesda: PAP IIID1 = LSIL; PAP IIID2 = HSIL; PAP III-p = ASC-H, PAP IVa-p = HSIL.

**Table 4 diagnostics-12-01748-t004:** Results for Pap IIID1, IIID2, III-p, and IVa-p in comparison the histologic results in women aged > 35.

Pap Smear (n = 1140)	hrHPV-Positive (n = 789)	hrHPV-Negative (n = 351)	Benign (n = 114)	CIN I (n = 329)	CIN II (n = 174)	CIN III/AIS (n = 504)	Carcinoma (n = 19)
Pap IIID1 (n = 316)	130 (41.1%)		14 (10.8%)	72 (55.4%)	25 (19.2%)	19 (14.6%)	0
Pap IIID1		186 (58.9%)	49 (26.3%)	120 (64.5%)	14 (7.5%)	3 (1.6%)	0
Pap IIID2 (n = 222)	162 (73.0%)		13 (8.0%)	21 (9.5%)	56 (25.2%)	72 (32.4%)	0
Pap IIID2		60 (27%)	5 (8.3%)	35 (58.3%)	16 (26.7%)	4 (6.7%)	0
Pap III-p (n = 158)	91 (57.6%)		15 (9.5%)	16 (17.6%)	11 (12.1%)	47 (51.6%)	2 (2.2%)
Pap III-p		67 (42.4%)	10 (14.9%)	40 (59.7%)	10 (14.9%)	7 (10.4%)	0
Pap IVa-p (n = 444)	406 (91.4%)		6 (1.5%)	21 (5.2%)	36 (8.1%)	327 (80.5%)	16 (3.9%)
Pap IVa-p		38 (8.6%)	2 (5.3%)	4 (10.5%)	6 (15.8%)	25 (65.8%)	1 (2.6%)

Abbreviations: AIS, adenocarcinoma in situ; CIN, cervical intraepithelial neoplasia; hrHPV, high-risk human papillomavirus. Comparison of Munich III and Bethesda: PAP IIID1 = LSIL; PAP IIID2 = HSIL; PAP III-p = ASC-H, PAP IVa-p = HSIL.

**Table 5 diagnostics-12-01748-t005:** Generalized Estimating Equation (GEE) Model with HPV and a Positive HPV Co-test as an Independent Predictor.

Pap (Reference Benign)	OR	95% CI	*p*
LSIL	2.52	1.765; 3.604	<0.001
HSIL+	34.27	24.629; 47.671	<0.001
Unspecific	9.24	6.346; 13.46	<0.001
HPV (reference: negative HPV co-test)			
Positive HPV co-test	5.07	4.068; 6.329	<0.001

Abbreviations: CI, confidence interval(s); HPV, human papillomavirus; HSIL, high-grade squamous intraepithelial lesion; LSIL, low-grade squamous intraepithelial lesion; OR, odds ratio.

## Data Availability

The data that support the findings of this study are available from the corresponding author upon reasonable request.

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
