# Peer review of "Cytology and High-Risk Human Papillomavirus Test for Cervical Cancer Screening Assessment"

_diagnostics, 2022, doi:10.3390/diagnostics12071748_

Round 1

Reviewer 1 Report

Stuebs and colleagues reported an original article on high-risk human papillomavirus (hrHPV) test coupled with citology test for cervical cancer screening assessment. Briefly, the authors conducted a retrospective analysis comparing the results of hrHPV and Pap co-test with the histological findings in each case seen in their certified dysplasia unit (January 2015 – October 2020) and found that a positive hrHPV co-test increased the risk for HSIL+ (5 fold). The manuscript is well written and well supported by consistent references. Overall, the obtained results encourage further studies to demonstrate the positive predictive value of cytology and hrHPV-based screening tests and the application of hrHPV test in routine clinical practice. However, I have some minor comments as follows:

1) Line 102, “oft he vagina und vulvar” must be correct into “of the vagina and vulva”.

2) Lines 112-117, The authors reported that HC2 test or Abbott RealTime high-risk HPV assay were used to detect hrHPV. Please indicate the number of hrHPV tests performed with each assay. Moreover, both tests should be better described in the Material and Methods section.

3) Did you have any inclusion and exclusion criteria for the enrolled subjects?

4) The results reported in Figure 1 should be better discuss in the main text.

Author Response

Reviewer 1

Stuebs and colleagues reported an original article on high-risk human papillomavirus (hrHPV) test coupled with cytology test for cervical cancer screening assessment. Briefly, the authors conducted a retrospective analysis comparing the results of hrHPV and Pap co-test with the histological findings in each case seen in their certified dysplasia unit (January 2015 – October 2020) and found that a positive hrHPV co-test increased the risk for HSIL+ (5 fold). The manuscript is well written and well supported by consistent references. Overall, the obtained results encourage further studies to demonstrate the positive predictive value of cytology and hrHPV-based screening tests and the application of hrHPV test in routine clinical practice. However, I have some minor comments as follows:

1) Line 102, “oft he vagina und vulvar” must be correct into “of the vagina and vulva”.

Thank you very much for pointing out this spelling mistake. This is due to the german autocorrection. We have corrected it.

2) Lines 112-117, The authors reported that HC2 test or Abbott RealTime high-risk HPV assay were used to detect hrHPV. Please indicate the number of hrHPV tests performed with each assay. Moreover, both tests should be better described in the Material and Methods section.

Thank you for this valid point. We have described the two tests in more detail and added two more references. Between January 2015 and October 2018 in total 2459 HC2 tests were performed, followed by 1129 Abbott RealTime high-risk HPV assays. These figures were added as well.

3) Did you have any inclusion and exclusion criteria for the enrolled subjects?

All women with a colposcopy of the cervix, a cytology, a hrHPV test and histology of the cervix at the same visit were included. Also women who had no biopsy taken during the colposcopy but had an operation performed (eg. Conization or hysterectomy) following the colposcopy were included. All women without any of the mentioned criteria were excluded. Also women referred to us for dysplasia of the vagina or vulva were excluded. We have added a new Flowchart (now Figure 1) to show who was included and excluded.

4) The results reported in Figure 1 should be better discuss in the main text

We have added some more information in the main text for Figure 2 (former Figure1).

Reviewer 2 Report

Since a new screening program was implemented in Germany in January 2020 and no data were available for women referred to certified dysplasia units for secondary screening, Stuebs et al., investigated combined testing with Papanicolaou smears and high-risk human papillomavirus (hrHPV) and compared the data with the histological findings. The authors found that the accuracy of Pap smears is comparable to the screening results and observed a 5-fold increment for highgrade squamous lesion (HSIL) after a positive hrHPV test, thus suggesting that colposcopy is necessary to properly diagnose HSIL+.

Overall, the manuscript is well written and organized and has potential clinical interest.

Minor comments:

-Material and Methods should include only the raw materials tools and subjects used in the investigation. I would suggest avoiding extensive description and preparing a table for Pap grades subdivision.

-Results: poor description is provided for this section. Tables would need to be reorganized, accompanied by proper legend, and information provided in a clearer manner. For example, the authors could use a flowchart showing exclusion patients, reorganized the data according HPV status or age distribution. The authors should better describe clinical findings vs histology of biopsy and present a direct comparison of the different classifications used in the study.

-Table 3: Only positive for HPV 16/18 or also for other high risk HPV genotype?

-Table 4: Any data are associated to “HPV (reference: negative HPV co-test)”. Please, justify.

-Attention should be paid to numbers, see for example, line 29, line 33, line 263, line 296 …etc.

Author Response

Reviewer 2:

Since a new screening program was implemented in Germany in January 2020 and no data were available for women referred to certified dysplasia units for secondary screening, Stuebs et al., investigated combined testing with Papanicolaou smears and high-risk human papillomavirus (hrHPV) and compared the data with the histological findings. The authors found that the accuracy of Pap smears is comparable to the screening results and observed a 5-fold increment for highgrade squamous lesion (HSIL) after a positive hrHPV test, thus suggesting that colposcopy is necessary to properly diagnose HSIL+.

Overall, the manuscript is well written and organized and has potential clinical interest.

Minor comments:

-Material and Methods should include only the raw materials tools and subjects used in the investigation. I would suggest avoiding extensive description and preparing a table for Pap grades subdivision.

Thank you very much for pointing out this valid comment. We shortened the material and methods section and focused on the main aspects. A new table for the PAP grade subdivisions was created (now table 1).

-Results: poor description is provided for this section. Tables would need to be reorganized, accompanied by proper legend, and information provided in a clearer manner. For example, the authors could use a flowchart showing exclusion patients, reorganized the data according HPV status or age distribution. The authors should better describe clinical findings vs histology of biopsy and present a direct comparison of the different classifications used in the study.

Thank you for this comment. The authors are well aware of the complexity of the tables. The Munich III nomenclature used in Germany is more complex than the commonly used Bethesda system in other parts of the world. In Germany HSIL are also subdivided into CIN 2 and CIN 3, in order of the better prognosis of CIN 2. Taking these aspects into account makes the tables complex. We have altered the legends of the tables as suggested by you to make it clearer what the tables are about. A flowchart was added to the manuscript (now Figure 1). The tables for age distribution (now Table 3 and Table 4) are reduced to PAP IIID1, IIID2, PAP III-p, PAP IVa-p because of the complexity of the provided information. Hence this study was conducted in Germany the classification used is Munich III. The Bethesda classification was also added to make the study accessible for an international audience. We hope that you agree with the changes.

-Table 3: Only positive for HPV 16/18 or also for other high risk HPV genotype?

Thank you for this comment. The data in all tables include women positive for HPV 16/18 and HPV other.

-Table 4: Any data are associated to “HPV (reference: negative HPV co-test)”. Please, justify.

We have specified the statistical analysis. We hope to meet your requested changes.

-Attention should be paid to numbers, see for example, line 29, line 33, line 263, line 296 …etc.

We carefully reviewed the numbers in the text and have applied necessary changes.